# Hyperbolic Embeddings in Sequential Self-Attention for Improved Next-Item Recommendations

## Abstract

In recent years, self-attentive sequential learning models have surpassed conventional collaborative filtering techniques in next-item recommendation tasks. However, Euclidean geometry utilized in these models may not be optimal for capturing a complex structure of behavioral data. Building on recent advances in the application of hyperbolic geometry to collaborative filtering tasks, we propose a novel approach that leverages hyperbolic geometry in the sequential learning setting. Our approach replaces final output of the Euclidean models with a linear predictor in the non-linear hyperbolic space. We experimentally demonstrate that under certain conditions hyperbolic models may simultaneously improve recommendation quality and gain representational capacity. We identify several determining factors that affect the results, which include the ability of a loss function to preserve hyperbolic structure and the general compatibility of data with hyperbolic geometry. For the latter, we propose an empirical approach based on Gromov delta-hyperbolicity estimation that allows categorizing datasets as either compatible or not.

## 1 Introduction

Contemporary recommender systems have greatly benefited from cross-disciplinary research, especially from leveraging the advancements in the natural language processing (NLP) field. These advancements range from adapting latent semantic indexing techniques for compact user and item representations to the current use of artificial neural network (ANN) models with sequential learning architectures. In particular, a significant progress was made in developing effective and practical sequential self-attention models based on transformer architectures. These architectures possess remarkable capabilities for efficiently and reliably extracting patterns from sequential data, leading to their ubiquitous integration into recommender systems tasks Fang et al. (2020).

One of the early and successful adaptations of such approach was the Self-Attentive Sequential Recommendation model (`SASRec`) introduced by Kang & McAuley (2018). `SASRec` combined the idea of self-attention with an asymmetric weighting scheme that respects the causal order of items in a sequence. Unlike previous generation of ANN approaches, which often struggled to compete with classical models in standard collaborative filtering setting Dacrema et al. (2021); Ludewig et al. (2021); Rendle et al. (2020), `SASRec` consistently outperformed both non-sequential and sequential learning competitors. This seminal work led to a line of research on improved sequential learning architecture designs that included, for example, new structural improvements Qiu et al. (2022) and more elaborate input processing schemes Zhou et al. (2020) to achieve better model expressiveness and improve generalization ability of the networks. Remarkably, it was recently shown that even in its standard formulation the quality of the model can be significantly improved by simply switching to a more appropriate loss function than in the original implementation Klenitskiy & Vasilev (2023); Petrov & Macdonald (2023).

On the other hand, some of the concepts that were initially investigated and tested in the NLP field remain underexplored in application to recommender systems. Starting from the works by Nickel & Kiela (2017; 2018), it was shown that hyperbolic geometry provides remarkable properties in terms of capturing complex hierarchical structures in language data and managing the curse of dimensionality. Later, the interest in application of hyperbolic spaces in machine learning has spread

to other domains as well, including computer vision Khrulkov et al. (2020) and recommender systems Mirvakhabova et al. (2020). In the latter case, however, the achievements remain limited primarily to standard collaborative filtering setting. As noted in Mirvakhabova et al. (2020), adopting hyperbolic geometry in a compatible way with complex architectures can be a challenging task. Hence, it poses an interesting question whether the previous achievements can be transferred to the new setting of sequential learning for the next item recommendation task.

Furthermore, the practical suitability of hyperbolic models for arbitrary types of datasets remains uncertain. We hypothesize that the accuracy of hyperbolic models can be significantly influenced by the characteristics of input data. Perhaps, the main characteristic in this regard is data "hyperbolicity", i.e., the extent to which a data manifold is compatible with hyperbolic geometry. Gromov $\delta$-hyperbolicity serves as a valuable metric for quantifying this property as it allows estimating the space curvature Mirvakhabova et al. (2020). An accurate estimation of this parameter and the corresponding influence on hyperbolic models performance has not been extensively explored in prior research. Motivated by this gap in knowledge, we put additional efforts into the analysis of hyperbolic compatibility at the data level. Overall, our contributions in this work are as follows.

- We propose a new architecture that combines the sequential self-attention with the hyperbolic classifier layer in the Poincaré ball. It improves recommendations quality and compresses learned representations on the datasets structurally compatible with the hyperbolic geometry.
- We identify two categories of data that exert contrasting effects on the capability of hyperbolic models to learn accurate representations and generate high-quality recommendations.
- Using Gromov $\delta$-hyperbolicity, we improve estimates of the hyperbolic space curvature on real datasets by linking the accuracy of estimation to the machine precision setting. We show that it has a pronounced effect on the quality of recommendations.

## 2 PROBLEM FORMULATION

Consider a problem of modeling interactions between a set of users $\mathcal{U}$ and a set of $N$ items $\mathcal{I} = \{i_k : k = 1, \ldots, N\}$. Each user $u \in \mathcal{U}$ can be represented as an ordered subset forming an item sequence $\mathcal{S}_u = \{i_{\pi_t^u} : t = 1, \ldots, |\mathcal{S}_u|\}$, where index $\pi_t^u$ preserves the relative order of items according to the user's consumption history. To deal with varying consumption history lengths, *the sequences are truncated to contain only $n$ the most recent items* for each user. Sequences initially containing fewer than $n$ items are padded with the reserved fixed token $i_{\text{pad}}$ to the left until the length is exactly $n$. The task is to learn a model parameterized by weights $\theta$ that will maximize the likelihood

$$P_\theta(i_{\pi_t^u} | i_{\pi_{t-n}^u}, \ldots, i_{\pi_{t-1}^u})$$

of predicting the true next item after seeing up to $n$ previous items. It is solved jointly for all $u \in \mathcal{U}$. We will build our solution based on the SASRec model Kang & McAuley (2018) (see Appendix A for details on SASRec architecture) with additional modifications at the final output layer. We aim to construct a linear classifier in the non-linear space that effectively solves the next item prediction problem using hyperbolic representations. In the original implementation, the prediction layer in the final output of the model provides the relevance of a target item $i \in \mathcal{I}$ as the next-item candidate at step $t$. It is defined via an inner product of the corresponding representations learned by the model:

$$r_\theta(i, t) = \langle \mathbf{M}_i, \mathbf{F}_t^{(b)} \rangle, \tag{1}$$

where $\mathbf{M}_i$ is an item $i$ embedding, and $\mathbf{F}_t^{(b)}$ is a user sequence embedding at step $t$ (obtained via $b$ sequential learning blocks of SASRec, see Appendix A). User index $u$ is omitted for brevity.

The learning task is to identify whether the target item will be of a positive class, turning into a binary classification problem. The logistic loss is used to learn the corresponding decision surfaces. The resulting SASRec's optimization objective is defined via the binary cross-entropy loss (BCE):

$$\mathcal{L}_{\text{BCE}}(\theta) = -\sum_{u \in \mathcal{U}} \left[ \sum_{t=2}^n \log p_\theta\left(i_{\pi_t^u}, t\right) + \sum_{j \in \mathcal{S}_{-u,t}} \log\left(1 - p_\theta\left(j, t\right)\right) \right], \tag{2}$$

where $p_\theta(i, t) = \sigma\left(r_\theta(i, t)\right)$ are the output logits that measure the probability of observing the true next item; $\sigma(x) = \frac{1}{1+e^{-x}}$ is a sigmoid function, and $\mathcal{S}_{-u,t} \subset \mathcal{I}$ is a sample of negative examples[1],

---

[1]Although originally, the authors of SASRec propose a more general formulation for subsampling of negative examples, our study finds that their actual implementation corresponds to the definition we provide here.

i.e., a subset of items the user $u$ *has not yet interacted with* by the moment $t$. In the original implementation and in our experiments, $\mathcal{S}_{-u,t}$ consists of a single randomly chosen element from the yet-unvisited-by-user part of the item catalog. *Our goal is to replace $r_\theta(i,t)$ and consequently $p_\theta(i,t)$ with their hyperbolic variants*, which can potentially increase the representation capabilities of the overall model and improve the quality of recommendations.

## 3 PROPOSED APPROACH

For the purposes of adopting the hyperbolic geometry, it is convenient to view equation 1 as a linear predictor in the standard classification task. Below we generalize this view using a particular model of the hyperbolic space based on the *d-dimensional Poincaré ball* $\mathcal{B}_c^d$ with constant negative curvature $-\kappa = 1/c^2$ and radius $1/\sqrt{c}$. This model provides a convenient representation and allows redefining most algebraic operations through the well-established gyrovector formalism (see Appendix B).

### 3.1 BINARY CLASSIFICATION IN POINCARÉ BALL

We start by defining an affine linear hyperplane $H_{\mathbf{a},b} = \left\{ \mathbf{x} \in \mathbb{R}^d : \langle \mathbf{a}, \mathbf{x} \rangle - b = 0 \right\}, \mathbf{a} \in \mathbb{R}^d \setminus \{\mathbf{0}\}$, $b \in \mathbb{R}$. The SASRec's prediction rule $r_\theta(i,t)$ from equation 1 can be viewed as a particular case, where linear bias term is discarded, i.e., $b = 0$. The binary decision surfaces are then simply defined by $N$ hyperplanes $H_{\mathbf{M}_i,0}$ containing the origin and constructed from the learnable item embeddings $\mathbf{M}$. As shown in Ganea et al. (2018), the idea of a linear hyperplane can be transferred to the the hyperbolic space through re-parametraization $\tilde{H}_{\mathbf{a},\mathbf{p}} = H_{\mathbf{a},\langle\mathbf{a},\mathbf{p}\rangle}$, where $\mathbf{p} \in \mathbb{R}^d$ is a new parameter that determines an offset of the hyperplane from the origin by redefining the bias as $b = \langle \mathbf{a}, \mathbf{p} \rangle$. Rewriting the definition of the hyperplane as $\tilde{H}_{\mathbf{a},\mathbf{p}} = \left\{ \mathbf{x} \in \mathbb{R}^d : \langle -\mathbf{p} + \mathbf{a}, \mathbf{x} \rangle = 0 \right\}$ makes it compatible with the hyperbolic space through the gyrovector formalism in the Poincaré ball model. The resulting hyperbolic hyperplane is then rendered as:

$$\tilde{H}_{\mathbf{a},\mathbf{p}}^c := \left\{ \mathbf{x} \in \mathcal{B}_c^d : \langle -\mathbf{p} \oplus_c \mathbf{x}, \mathbf{a} \rangle = 0 \right\}, \tag{3}$$

where $\mathbf{p} \in \mathcal{B}_c^d$, $\mathbf{a} \in T_\mathbf{p}\mathcal{B}_c^d \setminus \{\mathbf{0}\}$, and $\oplus_c$ is the so called Möbius addition defined on $\mathcal{B}_c^d$. By analogy with the Euclidean space, $\tilde{H}_{\mathbf{a},\mathbf{p}}^c$ can be viewed as *the union of images of all geodesics in $\mathcal{B}_c^d$ orthogonal to $\mathbf{a}$ and passing through the point defined by $\mathbf{p}$*. The resulting decision surfaces (Poincaré hyperplanes) can be viewed as sphere segments intersecting $\mathcal{B}_c^d$ and orthogonal to its boundary, see Fig. 1 as an illustration. In this new view, the decision rule from equation 1 generalizes to the following classification predictor in $\mathcal{B}_c^d$:

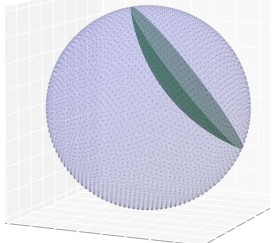

$$\tilde{v}_{\mathbf{a},\mathbf{p}}^c(\mathbf{x}) = \text{sign}\left(\langle -\mathbf{p} \oplus_c \mathbf{x}, \mathbf{a} \rangle\right) d_c\left(\mathbf{x}, \tilde{H}_{\mathbf{a},\mathbf{p}}^c\right) \|\mathbf{a}\|_\mathbf{p}^c, \tag{4}$$

where we rewrite $\tilde{v}_{\mathbf{a},\mathbf{p}}^c(\mathbf{x}) = \langle -\mathbf{p} \oplus_c \mathbf{x}, \mathbf{a} \rangle$ via the combination of the distance from a point to a hyperplane $d_c(\mathbf{x}, \tilde{H}_{\mathbf{a},\mathbf{p}})$ and the induced norm $\| \cdot \|_\mathbf{p}^c = \lambda_\mathbf{p}^c \| \cdot \|$. We refer the reader to Shimizu et al. (2021) for more details on this transformation.

Figure 1: Example of a decision surface in $\mathcal{B}_1^3$ represented by a hyperbolic hyperplane (green). It realizes a linear classifier in a non-linear space.

Note that equation 3 expands the number of parameters of the hyperplane from $n + 1$ to $2n$ due to the replacement of scalar $b$ with $n$-dimensional vector $\mathbf{p}$. This may lead to significant computational overhead, excessive memory usage, and an increased likelihood of overfitting the model. In this work, we adopt a more economic approach proposed by Shimizu et al. (2021). The authors highlight that the choice of $\mathbf{p}$ is arbitrary in the original definition of the hyperplane. Thus, by selecting $\mathbf{p}$ that points the same direction as $\mathbf{a}$, both can be represented via a single vector $\mathbf{z} \in T_\mathbf{0}\mathcal{B}_c^d \setminus \{\mathbf{0}\}$ defined in the tangent space $T_\mathbf{0}\mathcal{B}_c^d$ (isometric to the Euclidean) and starting from the origin. The only additional parameter needed is a scalar $r \in \mathbb{R}$ that scales $\mathbf{z}$ appropriately. The following procedure implements this re-parameterization via exponential mapping $\exp_\mathbf{0}^c$ and parallel transport $P_{\mathbf{0} \to \mathbf{p}}^c$ relative to the ball origin (see Appendix B):

$$\mathbf{p} = \exp_\mathbf{0}^c\left(r \frac{\mathbf{z}}{\|\mathbf{z}\|}\right), \quad \mathbf{a} = P_{\mathbf{0} \to \mathbf{p}}^c(\mathbf{z}), \tag{5}$$

i.e., a vector of length $r$ is exponentially mapped to the Poincaré ball in the direction defined by $\mathbf{z}$ that gives an offset of the Poincaré hyperplane. The vector $\mathbf{z}$ is additionally parallel-transported from

the origin to the offset point, fixing the hyperplane's orientation. The proposed method thus replaces the $(\mathbf{a}, \mathbf{p})$ parameterization with $(\mathbf{z}, r)$: $\tilde{H}_{\mathbf{a},\mathbf{p}}^c = \hat{H}_{\mathbf{z},r}^c$, $\tilde{v}_{\mathbf{a},\mathbf{p}}^c(\cdot) = \hat{v}_{\mathbf{z},r}^c(\cdot)$. Notably, *in the hyperbolic space, the offset may play a more significant role than in the Euclidean space*. For example, the further the hyperplane is from the origin, the easier it is to isolate classes (see e.g. (Doorenbos et al., 2023, Fig. 1)). In our preliminary experiments, the non-zero offset indeed slightly improved the results for the hyperbolic models, while no significant difference was observed in the Euclidean case.

Procedure in equation 5 establishes the generalization path of the SASRec's classification predictor to the hyperbolic geometry by defining $N$ independent Poincaré hyperplanes $\hat{H}_{\mathbf{z},r}^c$ for each $\mathbf{z} = \mathbf{M}_i$, $i = 1, \ldots, N$. The classifier decision is made over the final output of the self-attention blocks, which is also mapped to the Poincaré ball via the exponential mapping:

$$r_\theta^c(i, t) = \hat{v}_{\mathbf{M}_i, r}^c \left( \exp_{\mathbf{0}}^c \left( \mathbf{F}_t^{(b)} \right) \right). \qquad (6)$$

We emphasize that *the entire architecture remains the same and the only change is concerned with transformations in the prediction layer that only affect loss computation 2*. It employs hyperbolic representations of item sequence states $\mathbf{F}_t^{(b)}$ and item embeddings $\mathbf{M}_i$ by replacing the scalar product in 1 with the more general definition 3 that renders the form 6. We will call this model HSASRec.

### 3.2  MULTINOMIAL LOGISTIC REGRESSION IN POINCARÉ BALL

The objective defined in equation 2 serves as an approximation to the standard cross-entropy loss. The approximation allows mitigating the computational burden of generating a probability distribution over the entire item catalog at each step $t$. This is particularly useful when the catalog size is exceptionally large, reaching millions of items, as the resulting memory and computational overhead becomes unmanageable. However, in domains like movie recommendations, where the number of items is typically much smaller, this overhead can be handled efficiently with modern hardware.

More importantly, BCE loss depends on negative sampling, which is typically implemented via a uniform sampling of items from the non-yet-interacted part of an item catalog. From the learning perspective, *such sampling may not play well with non-Euclidean geometry as it may hinder the structural information viable for learning a proper data representation*. Indeed, as indicated by Krioukov et al. (2010), one of the most prominent features of the hyperbolic space is the ability to capture global properties in data. In recommender systems, it can be distributional properties of items based on their popularity hierarchy. The points representing items in the hyperbolic space will carry information about their position in this hierarchy. We hypothesize that *negative sampling may prevent the learning process from capturing the true structure of data*. As shown in Fig. 2, the distribution of negative samples is far from uniform, exhibiting a pronounced bias against popular items.

Hence, to verify our hypothesis, we run additional experiments where BCE is replaced by standard cross-entropy. The corresponding classification task turns into the multinomial logistic regression problem. The setting for this problem is straightforwardly achieved from the same reparametrization process in 5. The output distribution over an entire item catalog at step $t$ is obtained via

$$p_\theta^c(i, t) = \text{softmax}\left( r_\theta^c(i, t) \right), \qquad (7)$$

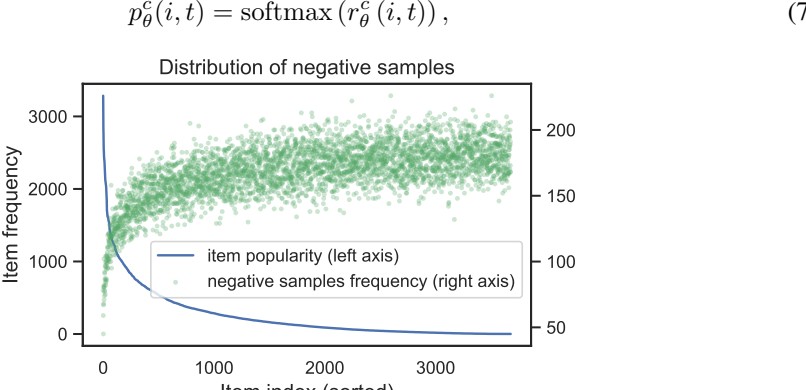

Figure 2: Uniform sampling of negative examples generates uneven distribution with the induced bias against popular items. This may ruin learning capabilities of models sensitive to popularity hierarchy and motivates the choice in favor of the CE loss. Movielens-1M dataset is used in this example.

where $\mathrm{softmax}$ acts over all items in $\mathcal{I}$ without sampling. The corresponding cross-entropy loss reads

$$\mathcal{L}_{\mathrm{CE}}(\theta) = -\sum_{u \in \mathcal{U}} \sum_{t=2}^{n} \log p_\theta^c \left( i_{\pi_t^u}, t \right). \tag{8}$$

We call the resulting model HSASRecCE. The CE part emphasizes the use of the cross-entropy loss.

### 3.3 SPACE CURVATURE ESTIMATION AND INEXACT ARITHMETIC

Space curvature $c$ may play a significant role in the accuracy of representations learned by a hyperbolic model. Indeed, if the space is strongly hyperbolic, then lower values of $c$ may limit the representation capacity of the model. Conversely, for overestimated $c$, the increased complexity may not bring any additional value. Assuming that the data lies on a hyperbolic manifold, its curvature is estimated via relation to the "ideal" hyperbolic space represented by the unit-radius Poincaré disk $\mathcal{B}_1^2$:

$$c(X) = \left( \frac{\delta_P}{\delta_X} \right)^2, \tag{9}$$

where $\delta_X$ is the Gromov's $\delta$-hyperbolicity of a dataset $X$ obtained via computation of the Gromov products Gromov (1987); $\delta_P = \ln(1 + \sqrt{2})$ is the $\delta$-hyperbolicity of the unit Poincaré disk Tifrea et al. (2018). In practice, a scale-invariant relative value $\delta_{rel}(X) = \frac{2\delta_X}{\mathrm{diam}(X)}$ is used, where $\mathrm{diam}(X)$ denotes a dataset diameter Borassi et al. (2015). Unlike raw $\delta_X$, it accounts for possible variations in data points spreading across the space. See Appendix C for details on the calculation.

Calculation of the relative $\delta$-hyperbolicity of the Poincaré disk $\delta_{rel}(P)$ is more intricate, though. Due to exponential expansion of the space closer to its boundary, certain operations become unreliable or even unavailable in the non-exact arithmetic. In particular, approaching the disk's boundary is limited by the machine precision $\epsilon$ and the estimates are obtained by setting the realistically achievable effective boundary that lies within the disk (see Fig. 3). The effective diameter of the unit Poincaré disk is therefore computed by estimating the outermost reachable boundary defined by $\epsilon$. The corresponding distance from the origin to the effective boundary in the Euclidean space is given by $r_E = 1 - \epsilon$, which yields the following estimation of the effective Poincaré disk radius:

$$r_P = 2 \tanh^{-1}(r_E) = \ln \frac{1 + r_E}{1 - r_E} = \ln \left( 1 + \frac{2 r_E}{\epsilon} \right). \tag{10}$$

The corresponding space diameter is calculated simply as $\mathrm{diam}(P) = 2 r_P$. The space curvature is then computed as:

$$c(X) = \left( \frac{\delta_{rel}(P)}{\delta_{rel}(X)} \right)^2. \tag{11}$$

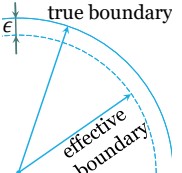

Figure 3: Illustration of an effective Poincaré disk boundary.

Many prior works Khrulkov et al. (2020); Mirvakhabova et al. (2020) used the default value of machine precision set to $\epsilon = 10^{-5}$. We critically examine this setting in our experiments. As demonstrated on Fig. 4 (left), changing $\epsilon$ has a dramatic impact on the curvature estimation, leading to an order of magnitude difference in the obtained values of $c$. Most importantly, *an improved estimation of the curvature leads to an improved accuracy of hyperbolic models*. As shown on Fig. 4 (right), there is a general trend of increased quality of recommendations with the increased machine precision. We empirically found $\epsilon = 10^{-12}$ to work well in most cases.

## 4 METHODOLOGY

We generally follow the experimental setup outlined in Kang & McAuley (2018). However, two modifications are made to better comply with the commonly accepted standards of evaluation Dacrema et al. (2021). Firstly, we disregard item catalog subsampling to score against the true hidden item. This subsampling yields inconsistent results Krichene & Rendle (2020). Secondly, we employ global timepoint splits for constructing train and test parts of the datasets. The scheme addresses the "recommendations from the future" issue Meng et al. (2020); Ji et al. (2023), which can lead to unfair evaluation. We evaluate the quality of recommendations using standard HitRate (HR) and Normalized Discounted Cumulative Gain (NDCG) metrics reported in Kang & McAuley (2018).

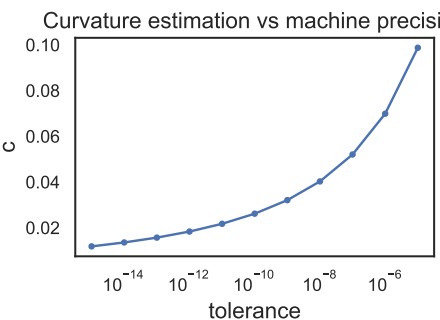
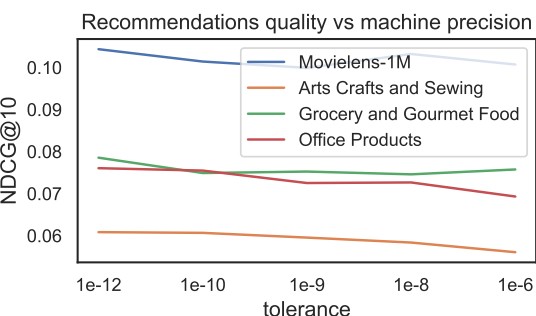

Figure 4: Illustration of an effect of machine tolerance $\epsilon$ on the space curvature estimation (left) and one the quality recommendations (right). The decreased machine precision used in space curvature estimation impedes the learning ability of the hyperbolic models.

We additionally report two more metrics: mean reciprocal rank (MRR) and item catalog coverage (COV). The latter quantifies the fraction of unique recommended items across all test users compared to the total number of unique items in the catalog. Lower values suggest that the algorithm tends to prioritize general patterns and lacks personalization. We conduct experiments on several publicly available datasets provided by Grouplens and Amazon. We only report results on four datasets most representative in terms of the revealed effects: Movielens-1M, Amazon Office Products, Amazon Grocery and Gourmet Food, and Arts Crafts and Sewing. The results on other datasets follow similar patterns and provide no new insights. See Appendix D for more details on datasets and evaluation.

### 4.1 BASELINES

As outlined in Section 5, significant efforts have been made in the development of novel sequential learning approaches in recent years. We, however, narrow our focus to one of the most straightforward sequential learning frameworks based on the initial `SASRec` architecture to analyze the fundamental aspects of applying hyperbolic geometry. This choice also echoes most recent endeavors on improving `SASRec` in the Euclidean space Klenitskiy & Vasilev (2023); Petrov & Macdonald (2023). We aim to isolate the effects of applying new geometry from the effects related to more elaborate data-preprocessing steps or complex architectures. We emphasize that our approach can be potentially applied to the newer architectures as well, preserving their base structure and translating the output to the hyperbolic space. However, constructing compatible hyperbolic output layers may present additional challenges and requires further research. Hence, we benchmark the following four `SASRec`-based architectures, all of which are implemented in PyTorch (Anonymous (2023)):

**SASRec.** The model implemented in Kang & McAuley (2018) serving as the Euclidean baseline.
**SASRecCE.** Another Euclidean model utilizing full cross-entropy loss instead of SASRec's BCE. Note that this approach presents a very strong baseline comparing even to more elaborate models such as BERT4Rec Sun et al. (2019). As demonstrated by Klenitskiy & Vasilev (2023), this variant of SASRec (which the authors call SASRec+ in their work) compares favorably to BERT4Rec on a wide selection of datasets. The authors also present a sampling mechanism to improve efficiency of the cross-entropy computation, which further improves the quality of recommendations. We restrict our study to the unsampled case due to the non-trivial effects of sampling on the learning ability of hyperbolic models, which we discuss in more details in Section 6.
**HSASRec.** A hyperbolic model that shares the same architecture as `SASRec`, except for the final layer that maps the output into the Poincaré ball before computing BCE loss.
**HSASRecCE.** Our main model that realizes a hyperbolic variant of SASRecCE in a similar fashion by mapping final output to the hyperbolic space before computing the CE loss.

As the hyperbolic models are expected to be more expressive, *we limit the allowed embedding size values to* $(32, 64, 128)$, *while the Euclidean models explore higher values from* $(256, 512, 728)$. Hence, we force the hyperbolic models to learn more compact representations. Other details on the values and ranges of hyper-parameters are provided in the Appendix E

For more comprehensive comparison, we additionally implement several standard baselines:
**PureSVD-N.** EigenRec-inspired Nikolakopoulos et al. (2019) version of PureSVD Cremonesi et al. (2010), that improves both recommendations' quality and diversity Frolov & Oseledets (2019).
**EASEr.** Steck (2019) is a linear model with additional constraints on the learned weights. It is one of the strongest baselines outperforming many classical and ANN-based models.

## 5 RELATED WORK

Several improvements over `SASRec` were made in recent years. Most of these improvements are *based on structural changes in the base architecture*. For example, MFGAN Ren et al. (2020) utilizes adversarial learning, featuring an attention-based generator and multiple dedicated discriminators. Locker He et al. (2021) combines local encoders with global attention heads to properly capture short-term user dynamics. STOSA Fan et al. (2022) employs Wasserstein Self-Attention module over stochastic Gaussian distribution of item embeddings, and a ranking loss regularization. DuoRec Qiu et al. (2022) incorporates contrastive regularization with additional masking augmentations and sampling strategies. Some works explore more efficient alternatives to the `SASRec`'s self-attention. C3SASR Chen et al. (2022) utilizes fast causal convolutions to create a lightweight sequential self-attention module. LASATF Frolov & Oseledets (2023) uses a non-neural approach based on hankel-matrix format and trains a low-rank tensor factorization model for the next-item prediction. Some other approaches *require additional input preprocessing* in order to train the model. For example, BERT4Rec Sun et al. (2019) uses bi-directional attention mechanism with additional masking and sampling of the input elements, aimed at improving the sequential learning ability. $S^3$-Rec Zhou et al. (2020) leverages self-attentive architecture for maximizing mutual information among items, their attributes, sequences, and sequential segments. LSSA Xu et al. (2021) separates user's short- and long-term preferences and applies dedicated self-attention layers to each representation.

In contrast to the aforementioned techniques, our approach offers a straightforward modification to `SASRec` that neither requires structural changes to the base architecture nor does it change an input to the model. Potentially, this approach can be applied to other architectures as well. This, however, requires additional analysis of the compatibility of neural network components and objective functions with the new geometry and goes beyond the scope of this work. As we demonstrate in Section 6, even as simple modification as a negative sampling in the loss may have a dramatic impact on the performance of hyperbolic models. In this context, the research most closely aligned by its spirit with the present work has been conducted by Klenitskiy & Vasilev (2023) and Petrov & Macdonald (2023). It demonstrates enhancements to the `SASRec` model through implementation of various cross-entropy loss functions and negative sampling strategies, albeit in the Euclidean space.

The research on hyperbolic recommender systems focuses primarily on the non-sequential models. For example, Chamberlain et al. (2019) use asymmetric (user-free) hyperbolic representations via Einstein midpoint-based aggregation, Vinh Tran et al. (2020) redefine the metric learning task in the hyperbolic space, Mirvakhabova et al. (2020) generalize autoencoders architectures. The graph-based approaches dominate more recent research. Sun et al. (2021) learn summarized representations over bipartite graph of interactions using the margin ranking loss with hyperbolic distances. HCGR Guo et al. (2021) is a contrastive learning model based on a hyperbolic graph architecture aimed at capturing hierarchical representations of items. Several models also attempted to combine graph representations with sequential learning. $H^2$SeqRec Li et al. (2021) utilizes a hyperbolic hypergraph convolutional neural network to learn sequential item embeddings. PHGR Guo et al. (2023) captures hierarchical information using a global heterogeneous and a local homogeneous graph representation learning. These two models are the most relevant to the task considered in the current work. However, it is generally difficult to separate the contribution of the hyperbolic geometry over graph-based representations to the final quality improvement. These models also have much higher architectural complexity compared to the approach presented here, which places them into a different family of methods. To the best of our knowledge, *our work is the first attempt to extend the sequential self-attention architecture with the hyperbolic output layer to address the next item prediction problem*.

## 6 RESULTS

We considered two dimensions in our analysis: the effect of applying hyperbolic geometry and the effect of negative sampling on the learning. *These experiments essentially provide a comprehensive ablation study of all the components of the proposed approach.* Notably, we observed significant improvements in the hyperbolic model without negative sampling, but this effect was primarily observed on "good" datasets. Here, we define "good" datasets as those that exhibit small variance in $\delta$-hyperbolicity estimation. However, on "bad" datasets with high variance in the estimation, we observed an opposite effect with hyperbolic models underperforming the baselines. In all experiments, the *negative sampling persistently inhibited the learning ability of the hyperbolic models*.

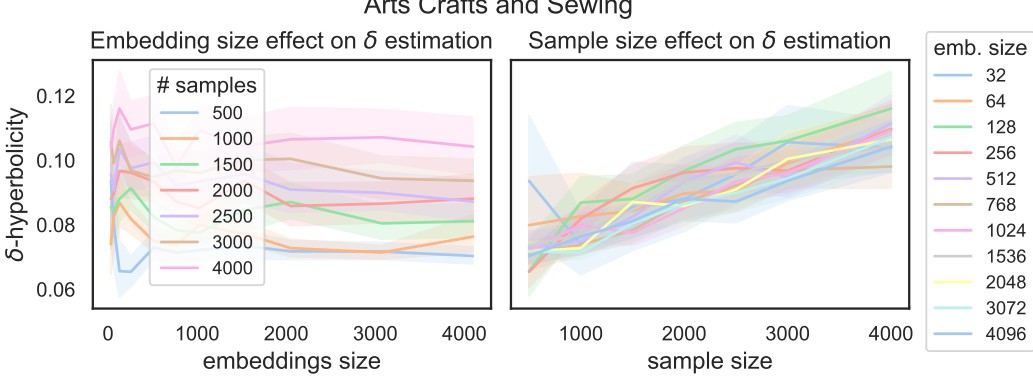

Figure 5: Estimation of $\delta$-hyperbolicity on a "good" dataset: Movielens-1M. As we increase both the sample size and the embedding size, the curvatures tend to converge to approximately the same value, indicated by higher concentration of curves on both graphs. More importantly, the values of $\delta$ exhibit a plateau (right figure) with increased sample sizes. It makes estimation of $\delta$ reliable.

Figure 6: Estimating $\delta$-hyperbolicity on a "bad" dataset: Amazon's Arts Crafts and Sewing. No convergence and no plateau (right figure) observed with the increased sample and embedding sizes, resulting in unreliable estimation of $c$, which hinders the ability of models to learn meaningful representations on such datasets.

## 6.1 DELTA ESTIMATION AND DATASETS CATEGORIES

Our results indicate that there is a certain pattern related to the estimation of the $\delta$-hyperbolicity of data. Some datasets exhibit stable behavior with respect to different sampling and SVD embedding sizes (obtained as described in Section C). For such datasets, the estimation of $\delta$ converges to approximately the same value, e.g., see Fig. 5. It gives a reliable estimate, which results in significant improvement of the recommendations quality when working with a hyperbolic model.

Some other datasets exhibit an opposite behavior, e.g., see Fig. 6. Estimation of $\delta$ on these datasets tends to have high variance, which makes it hard to properly select the most appropriate value. There is no plateau and the value of $\delta$ keeps growing even with large sample sizes. It indicates that the datasets do not align with hyperbolic manifolds. This, in turn, affects estimation of the space curvature $c$ and leads to inferior quality of the hyperbolic models independently of the settings.

Our experiments indicate that both *Movielens-1M* and *Grocery and Gourmet Food* datasets have a "good" internal structure that allows estimating $\delta$ more reliably, which results in better recommendations (see Table 1). Conversely, *Office Products* and *Arts Crafts and Sewing* demonstrate an opposite effect. On these datasets, the models tend to be Euclidean (increased $\delta$ means lower $c$). In our setting with the limited embeddings size (described in Section 4.1) it hampers their representation capacity.

## 6.2 RECOMMENDATIONS QUALITY

All evaluation results are provided in the Table 1, where the best values are highlighted in bold font and the second best are underlined. From now on we focus on the full HSASRecCE model.

Table 1: Evaluation results.

| Dataset | Metric | PureSVD-N | EASEr | SASRec | HSASRec | SASRecCE | HSASRecCE |
|---------|--------|-----------|-------|--------|---------|----------|-----------|
| Movielens-1M | NDCG@10 | 0.0313 | 0.0324 | 0.0802 | 0.0731 | 0.0847 | **0.0876** |
| | HR@10 | 0.0634 | 0.0655 | 0.1529 | 0.1427 | 0.1627 | **0.1685** |
| | MRR@10 | 0.0217 | 0.0225 | 0.0582 | 0.0522 | 0.0611 | **0.0631** |
| | COV@10 | 0.4873 | 0.3707 | 0.7386 | 0.6668 | **0.7768** | 0.7500 |
| Grocery and Gourmet Food | NDCG@10 | 0.0561 | 0.0656 | 0.0607 | 0.0527 | 0.0686 | **0.0696** |
| | HR@10 | 0.0801 | 0.0933 | 0.0840 | 0.0786 | 0.1007 | **0.1032** |
| | MRR@10 | 0.0485 | 0.0569 | 0.0533 | 0.0446 | 0.0586 | **0.0591** |
| | COV@10 | 0.3304 | 0.2750 | 0.0533 | 0.2516 | 0.2415 | **0.3882** |
| Office Products | NDCG@10 | 0.0530 | 0.0599 | **0.0648** | 0.0372 | 0.0629 | 0.0598 |
| | HR@10 | 0.0722 | 0.0826 | 0.0891 | 0.0588 | **0.0903** | 0.0854 |
| | MRR@10 | 0.0470 | 0.0528 | **0.0573** | 0.0306 | 0.0544 | 0.0520 |
| | COV@10 | **0.4636** | 0.2851 | 0.1957 | 0.1066 | 0.2016 | 0.4080 |
| Arts Crafts and Sewing | NDCG@10 | 0.0462 | 0.0531 | 0.0399 | 0.0386 | **0.0571** | 0.0520 |
| | HR@10 | 0.0642 | 0.0779 | 0.0624 | 0.0602 | **0.0877** | 0.0799 |
| | MRR@10 | 0.0406 | 0.0454 | 0.0330 | 0.0320 | **0.0478** | 0.0434 |
| | COV@10 | 0.5781 | 0.3684 | 0.1824 | 0.3572 | 0.3353 | **0.6228** |

Our experiments indicate that the Movielens-1M and Grocery and Gourmet Food datasets can be classified as "good", i.e., compatible with hyperbolic geometry. Notably, the quality metrics for these datasets surpassed the baseline models, demonstrating the superiority of the hyperbolic models in these cases. Conversely, the Office Products and Arts Crafts and Sewing datasets were identified as "bad" in terms of data hyperbolicity, and we observed poor performance on these datasets.

For "good" datasets, we observe significant improvements in recommendation quality when utilizing hyperbolic models. The quality improvement ranges from 8% to 18% compared to the original Euclidean baseline. This demonstrates the superiority of our hyperbolic models in accurately predicting user preferences and providing high-ranking recommendations. Furthermore, even when compared to the adjusted Euclidean baseline with full cross-entropy loss (SASRecCE), our hyperbolic models still demonstrate improvement, albeit to a lesser extent within the 3% range. An important factor for consideration though is that *our models use smaller embeddings size* as described in Appendix E. For example, on the Movielense dataset the optimal embeddings size for HSASRecCE is 32, while the optimal embeddings size for SASRec is 512 (see also Table 4 in Appendix F). This signifies that the hyperbolic models capture additional information and leverage the advantages of hyperbolic geometry by having more expressive and compact representations at the same time.

Finally, we highlight the overall poor performance of the HSASRec model. We attribute it to the negative sampling used in the loss computation. As demonstrated on Fig. 2, the sampling leads to a significant bias against the popularity of items. On the other hand, the popularity-based hierarchies are the best candidates to be properly captured by hyperbolic geometry. Hence, the HSASRec model suffers from the under-representation of input data and fails to learn viable relations within it.

## 7 CONCLUSIONS AND DISCUSSION

We implemented several hyperbolic models that learn separating hyperplanes within the Poincaré ball space. The models extend the SASRec architecture by only adjusting the output prediction layer. Essentially, our models incorporate a linear classifier in a non-linear hyperbolic space. Notably, the learnable models' weights remain Euclidean, eliminating the need for Riemannian optimization.

We demonstrated that our models based on the hyperbolic MLR layer consistently outperformed both the original `SASRec` baseline and the adjusted Euclidean baseline with the similar full cross-entropy loss. Remarkably, it was achieved with *considerably smaller embeddings size*, making the models more compact. These results demonstrate the effectiveness and superiority of hyperbolic models in enhancing recommendation quality. However, it is essential to acknowledge that the structural compatibility of data with hyperbolic geometry plays a crucial role in determining the extent of these improvements. For datasets that do not conform well with hyperbolic geometry, the benefits of using hyperbolic models may be limited or even negative comparing to the Euclidean ones. It remains an open question, though, whether the problem of compatibility with hyperbolic geometry arises from the nature of data or it attributes to space curvature estimation and can be fixed with better algorithms.

We also showed the adverse impact of negative sampling that impedes the ability of hyperbolic models to capture global data structure. It further emphasizes the importance of understanding the interplay between data characteristics and hyperbolic modeling in the quest for improved recommendations.

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

## A  REUSED PARTS OF THE SASREC ARCHITECTURE

Similarly to large language models, input items are encoded via an embedding matrix $\mathbf{M} \in \mathbb{R}^{N \times d}$ with the hidden dimension size $d$ in the Euclidean space. Consequently, each input user sequence is represented as an $n \times d$ matrix $\mathbf{E} = \left( \mathbf{M}_{i_{\pi_1}}, \ldots, \mathbf{M}_{i_{\pi_n}} \right)$ comprised of the corresponding rows of $\mathbf{M}$. Here and further in the text, we omit user index for brevity. The representation is also enriched with fixed positional embeddings in an additive manner, which we do not explicitly show here but use in the actual implementation exactly as in the original work by Kang & McAuley (2018). Following the standard definition of the neural attention block

$$\text{Attention}(\mathbf{Q}, \mathbf{K}, \mathbf{V}) = \text{softmax}\left( \frac{\mathbf{Q}\mathbf{K}^{\top}}{\sqrt{d}} \right) \mathbf{V}, \tag{12}$$

the sequential self-attention layer is then defined as

$$\mathbf{S} = \text{SA}(\mathbf{E}) = \text{Attention}\left( \mathbf{E}\mathbf{W}_Q, \mathbf{E}\mathbf{W}_K, \mathbf{E}\mathbf{W}_V \right), \tag{13}$$

where $d \times d$ matrices $\mathbf{W}_Q, \mathbf{W}_K, \mathbf{W}_V$ are learnable parameters of the model. An additional lower triangular mask is employed in the dot-product between $\mathbf{Q}$ and $\mathbf{K}$ of the self-attention *to enforce causality*. This mask restricts the attention mechanism to only attend to the elements that have already been processed, preventing the model from attending to future elements in the sequence and making it well-aligned with the next item prediction task.

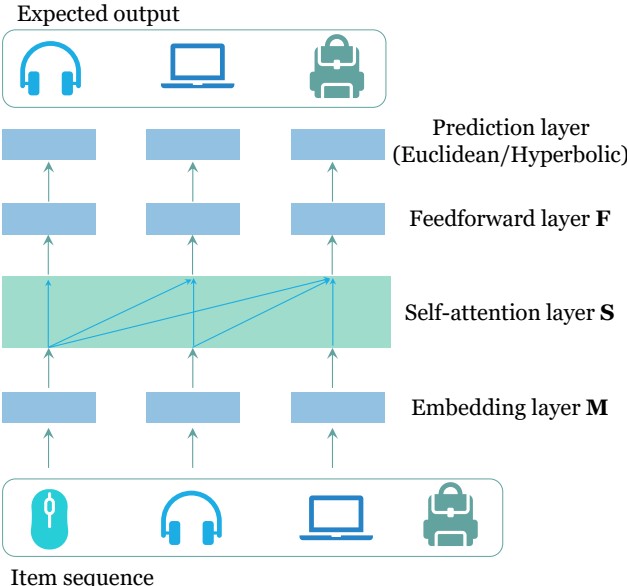

Figure 7: Schematic illustration of a sequential learning architecture based on self-attention. The output of the sequential learning block can be followed by either the Euclidean layer, which corresponds to the SASRec model, or the Hyperbolic layer that produces a new family of hyperbolic models.

The self-attention block is followed by a pointwise two-layer feed-forward network $\mathbf{F}$ with additional non-linearity in the form of ReLU. More precisely, each output state $\mathbf{S}_t$ of an input sequence at step $t$ is independently transformed into

$$\mathbf{F}_t = \text{FFN}(\mathbf{S}_t) = \mathbf{W}^{(2)} \text{ReLU}\left( \mathbf{W}^{(1)}\mathbf{S}_t + \mathbf{b}^{(1)} \right) + \mathbf{b}^{(2)}, \tag{14}$$

where $\mathbf{W}^{(1)}, \mathbf{W}^{(2)}$ are $d \times d$ matrices and $\mathbf{b}^{(1)}, \mathbf{b}^{(2)}$ are the bias vectors of the feed-forward network. The entire self-attention block transformation can be performed several times in a recursive manner:

$$\mathbf{F}_t^{(b)} = \text{FFN}\left( \mathbf{S}_t^{(b)} \right), \quad \forall t \in \{1, 2, \ldots, n\}, \tag{15}$$

$$\mathbf{S}^{(b)} = \text{SA}\left( \mathbf{F}^{(b-1)} \right), \tag{16}$$

where $b \geq 1$ is an integer hyper-parameter of the model controlling the number of stacked blocks. Initialization is done as $\mathbf{S}^{(1)} = \mathbf{S}$ and $\mathbf{F}^{(1)} = \mathbf{F}$. After each block, an additional three-fold transformation composed of layer normalization, dropout, and residual connection is applied independently for each step of the sequence as in the original `SASRec` model.

## B  HYPERBOLIC SPACE AND POINCARÉ BALL MODEL

The hyperbolic space $\mathbb{H}^d$ is a homogeneous, simply connected $d$-dimensional Riemannian manifold with constant negative sectional curvature $\kappa < 0$. The latter property makes the space analogous to the Euclidean sphere which has constant positive curvature. However, negative curvature induces fundamentally different geometrical properties which become especially useful for certain types of data and applications. While several isomorphic models of the hyperbolic space exist, in this paper, we will focus solely on exploring the Poincaré ball model denoted as $\mathcal{P}^d$. Below we provide a brief description and main definitions related to the Poincaré ball model. For a deeper dive into the topic, we refer the reader to Shimizu et al. (2021).

$\mathcal{P}^d$ is a model of hyperbolic space with constant negative curvature $-\kappa = 1/c^2$ realized by the $d$-dimensional open ball $\mathcal{B}_c^d = \{\mathbf{x} \in \mathbb{R}^d \colon c\|\mathbf{x}\|^2 < 1,\ c > 0\}$ with radius $1/\sqrt{c}$. The ball is equipped with the Riemannian metric tensor $g_{\mathcal{P}}(\mathbf{x}) = (\lambda_x^c)^2\, g_E$, where

$$\lambda_{\mathbf{x}}^c = \frac{2}{1 - c\|\mathbf{x}\|^2} \tag{17}$$

is the conformal factor and $g_E = \mathbf{I}_d$ denotes the Euclidean metric tensor. Here and throughout the text, by $\|\cdot\|$ we denote the ordinary Euclidean norm. The linear approximation of the hyperbolic space at any point $\mathbf{x} \in \mathcal{B}_c^d$ is denoted as $T_x\mathcal{B}_c^d$ and is called a *tangent space*. Conveniently, $T_x\mathcal{B}_c^d$ is isometric to the Euclidean space.

Poincaré ball of unit radius ($c = 1$) is often considered an "ideal" model of the hyperbolic space. However, real data may not always be ideally aligned with the hyperbolic manifold. Hence, varying values of $c$ allows adjusting for possible data incompatibility by bending hyperbolic space closer to or further away from the Euclidean geometry. For any fixed $c \neq 0$ the space expands exponentially as one moves closer to the boundary of the ball, while in the limit $c \to 0$ it recovers the Euclidean case.

**Algebraic setting**  The hyperbolic space can also be equipped with an algebraic framework based on the so called *gyrovector formalism* that provides a consistent analogy of natural operations in the Euclidean space. In particular, for any $\mathbf{x}, \mathbf{y} \in \mathcal{B}_c^d$, the hyperbolic (non-commutative) analog of the standard addition is called *Möbius addition* and is defined as:

$$\mathbf{x} \oplus_c \mathbf{y} := \frac{(1 + 2c\langle \mathbf{x}, \mathbf{y}\rangle + c\|\mathbf{y}\|^2)\mathbf{x} + (1 - c\|\mathbf{x}\|^2)\mathbf{y}}{1 + 2c\langle \mathbf{x}, \mathbf{y}\rangle + c^2\|\mathbf{x}\|^2\|\mathbf{y}\|^2}. \tag{18}$$

Subtraction is obtained by simply negating vector elements, i.e., $\mathbf{x} \oplus_c (-\mathbf{y})$. Note that $c = 0$ recovers the standard addition in $\mathbb{R}^d$. The induced distance between two points in the Poincaré ball is given by

$$d_c(\mathbf{x}, \mathbf{y}) = \frac{2}{\sqrt{c}} \tanh^{-1}\left(\sqrt{c}\,\|-\mathbf{x} \oplus_c \mathbf{y}\|\right). \tag{19}$$

We will also use it to define a distance from a point to a hyperbolic hyperplane introduced further in the text. In the limit $c \to 0$, equation 19 also recovers the Euclidean distance. Properties of these operations are extensively discussed in Shimizu et al. (2021). The Poincaré ball can also be equipped with exponential map $\exp_{\mathbf{x}}^c$, logarithmic map $\log_{\mathbf{x}}^c$, and Parallel transport $P_{\mathbf{x} \to \mathbf{y}}^c$. We are specifically interested in the following operations defined relative to the ball origin $\mathbf{x} = \mathbf{0}$:

$$\exp_{\mathbf{0}}^c(\mathbf{v}) = \tanh(\sqrt{c}\|\mathbf{v}\|)\frac{\mathbf{v}}{\sqrt{c}\|\mathbf{v}\|}, \quad P_{\mathbf{0} \to \mathbf{y}}^c(\mathbf{v}) = \frac{\lambda_{\mathbf{0}}^c}{\lambda_{\mathbf{y}}^c}\mathbf{v} \tag{20}$$

for any $\mathbf{v} \in T_{\mathbf{0}}\mathcal{B}_c^d$ and $\mathbf{y} \in \mathcal{B}_c^d$. Their use is explained in Section 3.

## C  COMPUTING DATA HYPERBOLICITY

Calculation of $\delta$-hyperbolicity has $O(n^4)$ complexity on the number of objects $n$, which can be problematic for datasets with millions of users and items. We therefore adopt an approximate scheme used in Mirvakhabova et al. (2020). The scheme is based on dimensionality reduction approach with

additional subsampling technique. We estimate $\delta$ value by first computing truncated SVD of user-item interactions matrix $\mathbf{A} = \mathbf{U}\boldsymbol{\Sigma}\mathbf{V}^{\top}$ with top $k$ leading singular values (i.e., $k$ defines embedding size). As our models do not explicitly represent users, we select PCA-style item representation $\mathbf{A}^{\top}\mathbf{U} = \mathbf{V}\boldsymbol{\Sigma}$ for further analysis. Based on that, we calculate $\delta$ in the corresponding item space using Gromov products that requires computing pairwise distances between the objects in the space. Since the number of items can be large, computing the full matrix of pairwise products can be intractable. Therefore we approximate $\delta$ by performing additional sampling to reduce the amount of objects in the calculation. We repeat the procedure several times with different subsamples and average the obtained values.

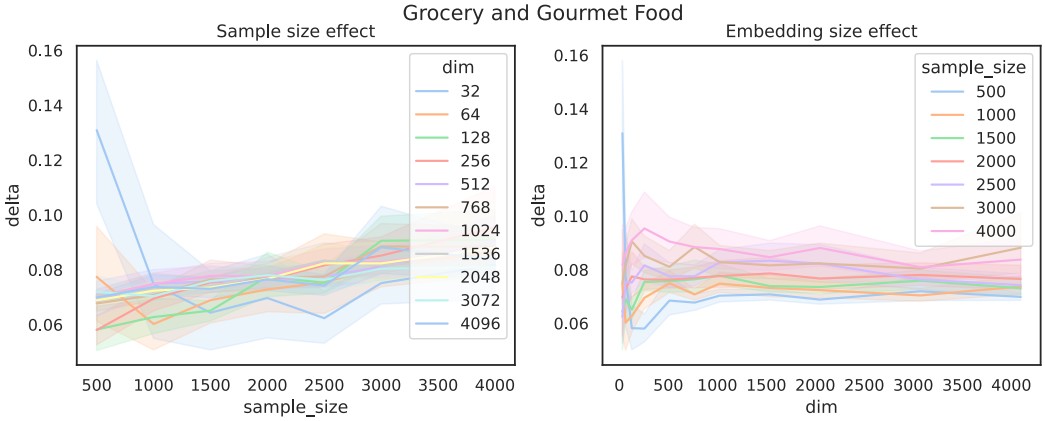

Figure 8: Delta hyperbolicity dependence on sample size and embedding size on Arts Crafts and Sewing dataset

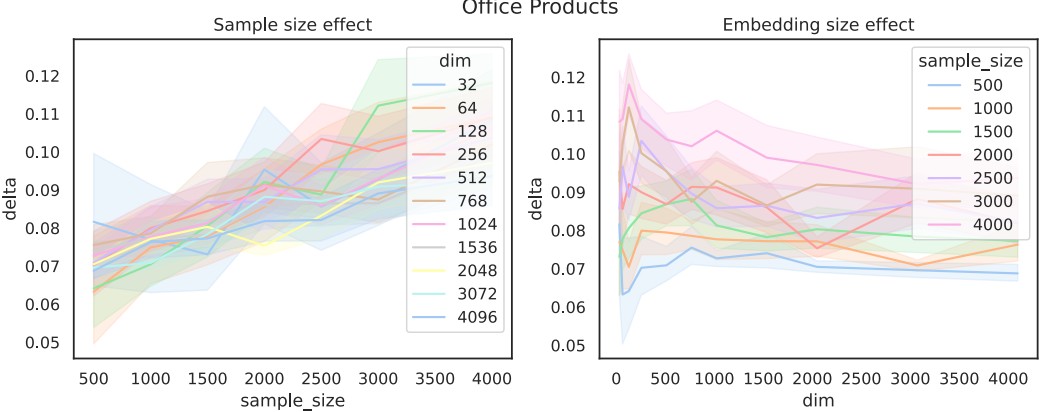

Figure 9: Delta hyperbolicity dependence on sample size and embedding size on Office Products dataset

## D    DATA PREPROCESSING AND EVALUATION DETAILS

All Amazon datasets are preprocessed using 5-core filtering that leaves no less than five interactions per each user and each item. The explicit values of ratings are disregarded, binarized values are used instead. Similarly to Kang & McAuley (2018), the maximum allowed length of user sequences is set to 200 for the Movielens dataset and to 50 on all Amazon datasets. Main dataset statistics are provided in Table 2.

We use global timepoint splits for evaluation of the models. Parts of users' histories before the timepoint become the training data. The remaining parts are used for evaluation of recommendations

Table 2: Main dataset statistics.

| Dataset | Users | Items | Interactions | Density |
|---|---|---|---|---|
| Arts Crafts and Sewing | 56210 | 22931 | 440318 | 0.00034 |
| Grocery and Gourmet Food | 127496 | 41320 | 1065424 | 0.0002 |
| Office Products | 101501 | 27965 | 740703 | 0.00026 |
| Movielens-1M | 6040 | 3706 | 1000209 | 0.04468 |

and are further split into the validation and final test parts using 2-5% of all interactions. We ensure that validation-test splits in evaluation contain approximately 5000-10000 interactions. The specific global time splitting intervals are as follows: four months for each split in the ML-1M case, three weeks for each split in the AMZ-B case, six weeks in the AMZ-G case, and two days for the test split and one day for validation in the Steam dataset. Hyper-parameter tuning is performed using the validation split. The optimal configuration is then used to retrain models on the combined training and validation parts and the final evaluation is conducted on the test split.

Following the global timepoint-split, we employ the successive evaluation scheme Fig. 10. At the beginning, all test users have two temporarily-split sequences: the sequence of already visited items, and the target sequence of several next items that were hidden at the training stage. For every test user, a model is requested to provide top-$n$ recommendations based on the current sequence of seen items. These recommendations are compared against the single true next item located in the beginning of the target sequence. The corresponding evaluation metrics are calculated and recorded. Then the true next item is excluded from the target sequence and appended to the end of the sequence of visited items. The evaluation process is repeated with these new sequences and continues until the target sequence is exhausted.

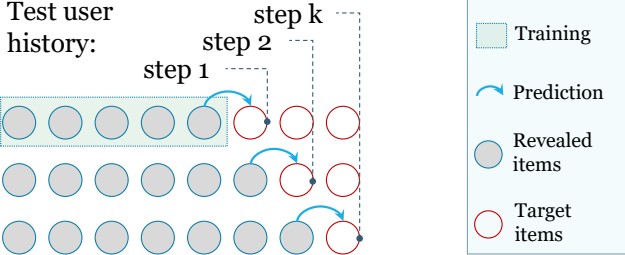

Figure 10: Illustration of the successive evaluation scheme for a single test user. Each new row represents the same user's history but different split into visited and target items. The recommendations are generated based on the user's current history of visited items. At each new step, the history is extended by appending the next target item until the full sequence is exhausted.

Due to successive evaluation scheme, we do not group scores by test users and perform calculations on a per-interaction basis. For example, the HR metric is calculated as the fraction of the total number of correct recommendations in the the total number of test interactions. If a user appears several times in the test split, we combine the hidden items from the previous test interactions with the user's training history in order to predict the hidden item at the current step. The history is always time-sorted, which ensures the forward direction of these steps in time.

## E  HYPER-PARAMETERS TUNING

For the PureSVD-based models we tune the rank of SVD and the popularity-based normalization factor $s$. The range of rank values $k$ is $(32, \ldots, 2048)$ with step size gradually increasing as powers of 2. For scaling $s$, the explored range is $(0.0, \ldots, 1.0)$ with step size $0.2$. The EASEr model only requires regularization factor for tuning. We take the corresponding values from the range $(8, \ldots, 3072)$ with values in the range defined by powers of 2. These models are trained on a single 48-core node with Intel Xeon Gold CPU @2.8GHz.

In the case of `SASRec`-based models, we explore values of the suggested hyper-parameters within the following ranges – batch size: $(64, 128, 256, 512)$, learning rate: $(0.00001, 0.0001, 0.001)$, the number of attention blocks: $(1, 2, 3)$, and dropout rate: $(0.2, 0.4, 0.6)$. The models are trained on NVidia A100 GPU with 80Gb RAM.

## F  COMPUTATIONAL ASPECTS

Hyperbolic models add additional overhead in the training time (Table 3), but the inference time remains almost the same (Fig. 11). The models also enable more compact representations as indicated by the learned optimal embedding size (Table 4).

Table 3: Training times comparison on the fixed set of hyper-parameters: number of epochs=20, batch size=256, learning rate=0.005, embedding size=32, number of blocks=3, number of attention heads=1, dropout rate=0.2.

| Dataset name | HSASRecCE | HSASRec | SASRec | SASRecCE |
|---|---|---|---|---|
| Movielens-1M | 52s | 34s | 22s | 28s |
| Grocery Gourmet Food | 27m:30s | 03m:35s | 02m:48s | 13m:23s |
| Arts Crafts and Sewing | 07m:08s | 01m:55s | 01m:25s | 03m:33s |
| Office Products | 15m:00s | 02m:49s | 02m:07s | 07m:18s |

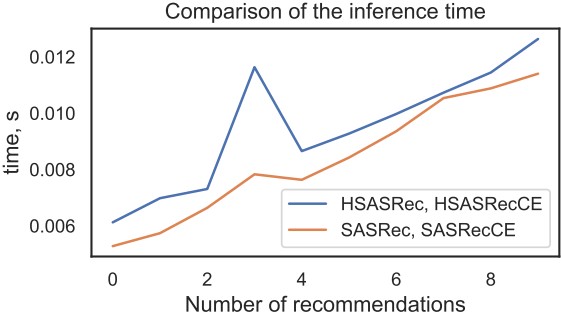

Figure 11: Inference time comparison

Table 4: Optimal size of the learned item embeddings.

| Dataset | SASRec | SASRecCE | HSASRecCE |
|---|---|---|---|
| Movielens-1M | 512 | 128 | **32** |
| Arts Crafts and Sewing | 256 | 256 | **32** |
| Grocery Gourmet Food | 256 | 128 | **64** |
| Office Products | 256 | 256 | **32** |

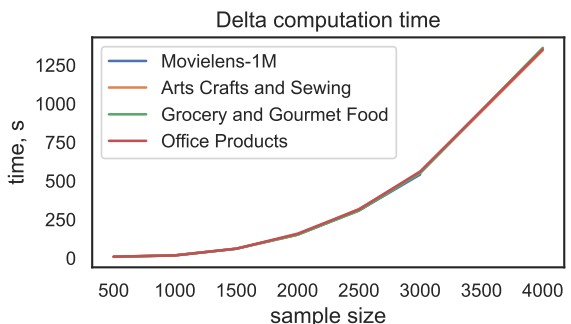

Figure 12: The time required for estimating $\delta$-hyperbolicity.

