# OpenReview forum: "Hyperbolic Embeddings in Sequential Self-Attention for Improved Next-Item Recommendations"
_ICLR.cc/2024/Conference — Submitted to ICLR 2024_

### Official Review · Reviewer_hnY1 · 2023-10-31

**Soundness:** 3 good
**Presentation:** 3 good
**Contribution:** 3 good
**Rating:** 5
**Confidence:** 3

**Summary:**

The paper makes the following main contributions:
- The paper proposes a new approach for the next-item recommendation that combines sequential self-attention with hyperbolic geometry. The base architecture is SASRec, with modifications only to the final prediction layer.

- The prediction layer is adapted to learn how to separate hyperplanes in the Poincaré ball, enabling a linear classifier in this non-linear hyperbolic space. This allows the model to leverage the benefits of hyperbolic geometry, like hierarchical representations and dimensionality reduction.

- An approach to estimate the hyperbolicity of datasets using Gromov delta-hyperbolicity is presented. Datasets are categorized as "good" or "bad" for hyperbolic modeling based on this.

**Strengths:**

The proposed approach is straightforward to implement, requiring only changes to the prediction layer of SASRec. This makes adoption more practical.

Analysis of dataset hyperbolicity provides insights into when these models can be expected to work well or not. The categorization into "good" vs "bad" datasets is useful.

**Weaknesses:**

There is no ablation study on the effects of different space curvature values. Varying the curvature and linking performance to estimated dataset hyperbolicity could provide better insights.

The negative sampling analysis seems incomplete. Different sampling strategies besides uniform should be evaluated before concluding their effects.

The approach for estimating dataset hyperbolicity lacks analysis of computational complexity and scalability. This could limit practical applications.

**Questions:**

Please solve the weakness listed above.

---

> ### Author Response · Authors · 2023-11-20
> **Response to Reviewer hnY1**
>
> Thank you for taking the time to thoroughly assess our work and provide valuable feedback. Below we address the concerns related to the listed weakness points. We quote the original point and then provide the details that should address it. We will separate the points into several subsequent comments due to the character limit per single comment.
>
> ### Point 1
> > There is no ablation study on the effects of different space curvature values. Varying the curvature and linking performance to estimated dataset hyperbolicity could provide better insights.
>
> While we do not dedicate a separate section to such analysis, we do show the effects of varying the space curvature on the performance. It is depicted in Figure 4. The graph on the left side illustrates the relationship between curvature values and machine precision. It demonstrates that the estimated space curvature values exhibit a considerable range of variation across different machine precision settings. The right part of the graph links the performance of the hyperbolic model to the same machine precision settings. Combining both parts indicates the following trend: higher precision aligns with lower values of $c$ and subsequently yields improved recommendation quality. While this figure was meant to demonstrate the importance of accurately estimating the values of $c$ (which is one of our contributions), it also demonstrates that overestimating these values may degrade the performance and provides some insights into the overall performance of hyperbolic models on different datasets.
>
> In this context, it is also important to note that some prior works, e.g., Khrulkov et al. (2020), did not use the estimated values of $c$ in the final experiments, as they found lower values to provide better results (see the last paragraph of Section 3.1 in their work). According to our calculations, using a better estimation procedure proposed in our work would provide a much better estimate of $c$ that would be much closer to the optimal one (which is an order of magnitude lower) used by the authors in their final experiments.

---

> > ### Author Response · Authors · 2023-11-20
> > **Response to Reviewer hnY1 (continued)**
> >
> > ### Point 2
> > > The negative sampling analysis seems incomplete. Different sampling strategies besides uniform should be evaluated before concluding their effects.
> >
> > We acknowledge the potential value of exploring the sampling strategies effects, as suggested. In this work, our primary focus was directed towards comparing our approach against stronger baselines, particularly those identified in the recent findings of Klenitskiy & Vasilev (2023). These models employed the full CE loss over the standard SASRec architecture without the negative sampling and demonstrated their superiority over the sampled approach. We build upon these results aiming to explore further limits of quality improvements with non-Euclidean geometry in the full CE setting. The BCE-based models mostly serve to contrast the effects of applying a different geometry and the study of the effects of sampling was not the main target. We note that no prior work has been done in the direction of understanding the effects if negative sampling on the landscape of loss functions in the Hyperbolic space. Finding structural connections between the non-uniform negative sampling and the geometric properties of data requires significant efforts, which in our opinion deserves a dedicated investigation.
> >
> > ### Point 3
> > > The approach for estimating dataset hyperbolicity lacks analysis of computational complexity and scalability. This could limit practical applications.
> >
> > The complexity of estimating data-hyperbolicity is indeed an important topic. The worst-case complexity is known to be $O(n^4)$ in the number of objects $n$. We provide the corresponding numerical results in Figure 12 in the Appendix. It can be seen that for reasonable values of sample sizes (e.g., for which the estimation of $\delta$ gives reliable results at least on "good" datasets), the time needed to compute $\delta$ remains comparatively small (comparing to the time that would be otherwise required for the gird-search).
> >
> > The high complexity of the algorithm prohibits using it in the brute-force regime for entire space and prior works, e.g., Khrulkov et al. (2020), use subsampling to construct smaller space and circumvent the problem, which seems to work well in practice. We followed the same procedure in our work. As a side note, devising new algorithms with a better asymptotic is an interesting direction of research, which we plan to explore in our further work.
> >
> > Additionally, it's worth highlighting the effectiveness of space subsampling in estimating hyperbolicity on 'good' datasets like Movielens. Figure 5 illustrates that on such datasets, the estimated values quickly reach a plateau, requiring a significantly smaller subsample size for reliable $\delta$ estimation compared to the entire item space. However, this efficiency doesn't extend to "bad" datasets, as observed in the absence of a plateau within a reasonable subsample size range. Coupled with the fact that the value of $\delta$ continues growing on such datasets (and by definition it can only grow, never decrease), we can make a conclusion that further attempts to improve the estimate (e.g., continue increasing sample sizes) has diminishing returns. Indeed, as higher values of $\delta$ correspond to lower values of space curvature c, i.e., the space becomes closer to Euclidean and the most remarkable advantages of the hyperbolic models disappear.

---

### Official Review · Reviewer_WdM2 · 2023-11-02

**Soundness:** 3 good
**Presentation:** 3 good
**Contribution:** 2 fair
**Rating:** 3
**Confidence:** 5

**Summary:**

This paper extends SASRec [1] in hyperbolic space by adjusting the output prediction layer for next-item recommendation task.

[1] Self-attentive sequential recommendation. ICDM 2018.
[2] HME: A Hyperbolic Metric Embedding Approach for Next-POI Recommendation. SIGIR 2020.

**Strengths:**

Strengths:

-	The approach is straightforward and well-explained
-	The writing is clear and on point
-	The authors conducted experiments on various datasets for benchmarking

**Weaknesses:**

Weaknesses:

-	In my opinion, the contribution is marginal while the only change is the prediction layer (Eqn (6))
-	In Section 4.1, the authors mentioned that “we limit the allowed embedding size values to (32, 64, 128), while the Euclidean models explore higher values from (256, 512, 728)”. More deeper analyses would make the paper stronger
-	In Section 6.2, Table 1 shows that hyperbolic based solutions do not always show remarkable performance. It would be better if the authors also dive deeper into the details of the datasets / models, and evaluate about which scenarios can make hyperbolic based methods perform the best.
-	Ablation studies should be further given. For example, can we generate visualization for user / item embeddings to observe the ‘before’ and ‘after’ changing the prediction layer?
-	Missing citations / baselines such as [2]

Overall, more works need to be done.

[1] Self-attentive sequential recommendation. ICDM 2018.
[2] HME: A Hyperbolic Metric Embedding Approach for Next-POI Recommendation. SIGIR 2020.

**Questions:**

Please see my comments above

---

> ### Author Response · Authors · 2023-11-20
> **Response to Reviewer WdM2**
>
> We appreciate the reviewer's dedicated efforts in evaluating our work and providing actionable suggestions and feedback. We will address the concerns by quoting them and expanding the explanation below the quote. Due to space constraints, our response will be split into several comments below.
>
> ### Point 1
> > In my opinion, the contribution is marginal while the only change is the prediction layer (Eqn (6))
>
> While we do understand this concern, we respectfully emphasize that the final results are easier to judge once they're obtained. However, even seemingly simple solutions do not necessarily have an easy path to them. In designing our approach, we experimented with several very different architectures, none of which resulted in any improvement, expect the one presented in the paper.  The possibility to boost models performance be simply replacing the final layer with a linear classifier in a non-linear space provides a straightforward practical advantage. However, this is not the only contribution.
>
> We also provide several insights on the conceptual side. We highlight the non-trivial aspects of understanding and modeling the geometry of data, which remain underexplored in the machine learning literature. We showcase the compatibility issue of data with the hyperbolic space geometry and propose a new view on the problem of estimating the curvature of the hyperbolic space. The latter may become helpful in early indication of possible compatibility issues, replacing the need for running expensive experiments. We also expose the non-trivial effects related to the negative sampling in the case of hyperbolic models, which leads to a significant degradation of quality even with respect to the baseline Euclidean model (SASRec). To the best of our knowledge, such effects remain unknown to the community. We believe it layouts a new direction of research that may spark new theoretical results and lead to better understanding of the limitations and applicability of geometric approaches to real-world problems.
>
> ### Point 2
> > In Section 4.1, the authors mentioned that “we limit the allowed embedding size values to (32, 64, 128), while the Euclidean models explore higher values from (256, 512, 728)”. More deeper analyses would make the paper stronger
>
> We understand that the choice of ranges of the embedding size may seem non-standard for the typical experiments in the Euclidean space. However, one of the main promises of the Hyperbolic space is its higher representation capacity, starting from the known theoretical fact that any tree can be embedded into the two-dimensional hyperbolic space (Poincaré disk). While this theoretical bound is hard to achieve in pratice due to hardware's finite arithmetic precision, one may still expect to see a representation capacity improvement (lower embedding sizes) in the hyperbolic models over their Euclidean counterparts. We, however, show in our work that it only happens if data is geometrically-compatible. The choice of the reduced embedding sizes helps demonstrating that effect. If we lift the restriction on the smaller embedding size, then even on "bad" datasets the hyperbolic models can be trivially made to perform at the same level as their Euclidean counterparts. It simply takes setting higher values of $\delta$ (lower curvature $c$ respectively) -- in this case the hyperbolic models become practically indistinguishable from the Euclidean ones. This result, however, would prevent us from demonstrating the true behavior of hyperbolic models on problematic datasets, while making the formal results (quality metrics) look much better in comparison. We believe that this result would be too trivial to show and instead focus on the discrepancies in learning abilities depending on the geometric properties of data. We believe that our approach offers a more equitable perspective, helping to more effectively unveil the underlying intricacies associated with training hyperbolic models.

---

> > ### Author Response · Authors · 2023-11-20
> > **Response to Reviewer WdM2 (continued)**
> >
> > ### Point 3
> >
> > > In Section 6.2, Table 1 shows that hyperbolic based solutions do not always show remarkable performance. It would be better if the authors also dive deeper into the details of the datasets / models, and evaluate about which scenarios can make hyperbolic based methods perform the best.
> >
> > We acknowledge the need for a deeper analysis of the connection between hyperbolicity and the structural properties of data/models.
> >
> > Let us firstly address the _"hyperbolic based solutions do not always show remarkable performance"_ concern. As we already stated in the comment above, the hyperbolic models can show at minimum a comparable performance even on "bad" datasets. We do not show it in the results due to a trivial explanation. In our experiments, we deliberately restrict the embedding sizes of the hyperbolic models to be smaller than in the Euclidean case (smaller embedding size is one of the main promises of a better representational capacity offered by the hyperbolic geometry). So the poor performance of the models on "bad" datasets can be attributed to the data being not well-aligned with the hyperbolic structure. This, in turn, prevents the hyperbolic models from fully capitalizing on the superior representational capacity of the hyperbolic space, which is additionally severed by the limitations imposed on the embedding size. The models can be "forced" to perform on par with their Euclidean counterparts by lifting the small embedding size restriction. However, this approach offers limited insights as it essentially leads to a comparison between two nearly identical models. With high $\delta$ / low $c$ values, the hyperbolic model converges toward the characteristics of the Euclidean model, diminishing the distinction between the two.
> >
> > Let us now address the general concern on the deeper analysis. In terms of model design, we conducted an initial elaborate investigation (details of which are not included in this work) encompassing various architectures, among which only the one presented in the current work showed the potential for notable improvements over the Euclidean baselines. To the best of our knowledge, there is no unified theory that enables constructive insights into the development of neural networks solutions based on arbitrary architectures in hyperbolic geometry. The seminal work by Krioukov et al. (2010) only outlines the basic principles of utilizing hyperbolic geometry, with the subsequent works focusing on different aspects related to particular choices of a neural architectures. Hence, most of the investigations largely remain empirical. Furthermore, the potential for generalization from a collection of studies utilizing different architectures remains unclear. Due to much larger variability in the definitions of key algebraic operations and building blocks of hyperbolic neural networks, such an analysis presents a substantial challenge, which limits the extent to which such an investigation can be performed within a single conference paper.
> >
> > Regarding the data aspect, the key challenge lies in establishing connections between the geometric properties and the internal structure of the data. Apparently, it must go beyond the $\delta$-hyperbolicity estimation, as demonstrated in our work. It is non-obvious, though, which form such an endeavor should take, which properties of datasets (statistical, topological, etc.) must be explored in connection to data geometry. To the best of our knowledge, we were the first to highlight the potential issues in estimating space curvature on arbitrary collaborative filtering datasets. The true reasons of observing the "bad" datasets remain largely undiscovered and require further attention from the community. With this work we hope to bring the attention of the community to the outlined challenges.

---

> > > ### Author Response · Authors · 2023-11-20
> > > **Response to Reviewer WdM2 (continued)**
> > >
> > > ### Point 4
> > >
> > > > Ablation studies should be further given. For example, can we generate visualization for user / item embeddings to observe the ‘before’ and ‘after’ changing the prediction layer?
> > >
> > > We agree that visualization may help in better understanding of the general effects. Unfortunately, this task is not as straightforward as we'd like it to be. If a $d$-dimensional model is learned with $d$>2, one have to use additional dimensionality reduction techniques. We were unable to preserve the structure of hyperbolic embeddings space in this case, which makes the analysis non-informative. Coversely, with $d$=2, the models (both Euclidean and Hyperbolic) fail to learn meaningful representations, which makes performing any analysis obsolete.
> > >
> > > We also note that our experiments present the full ablation study of the architecture, as we test all main components independently in a separate model: Euclidean model with BCE, Euclidean model with CE, Hyperbolic model with BCE, Hyperbolic model with CE. Our experiments encompass all variations in the architecture. This helps us to draw informed conclusion on the effects related to both hyperbolic models efficiency (including the effects on negative sampling, mentioned above), and data compatibility.
> > >
> > > ### Point 5
> > >
> > > > Missing citations / baselines such as [2]
> > >
> > > We thank the reviewer for pointing out the citation. While this work indeed can be viewed as related, there are several major concerns related to considering it as a baseline. The architecture of the proposed solution is not sequential, it is a general learning scheme. What makes it suitable for the "next-POI recommendation" is the data preprocessing, which builds simple item-to-item transition statistics. Therefore, the model implements a first-order Markov chain mechanism and is not fully sequential in the modern sequential attention sense. It learns pairwise statistics and doesn't consider an entire item sequence as a whole object with internal ordered structure. In addition to that, unfortunately, the authors do not compare their approach to the baseline SASRec model. While the authors claim that their approach is suitable for next-item recommendation tasks, there are no experiments on datasets like Movielens or Amazon that are traditionally used to test against sequential learning scenario. All these aspects make it hard to see if their aproach can serve as a good baseline. Finally, there is no link to an open-source implementation of the proposed approach. We believe that sharing the source code with hyperbolic models is even more important than in the Euclidean case, as there are much more intricacies related to implementation (e.g., more numerical instabilities, different implementations of the operations in the hyperbolic space, etc.). We believe that fair comparison is only possible when the authors provide a fully-runnable source code of their solution.
> > >
> > > Finally, we noted the visualization of embeddings in [2], which indeed look nice. We hypothesize that such vivid structure is possible due to simpler objects encoded into the space (categories vs items). This does not seem to be an option in our general sequential learning setup, where the embedding space is less structured (no well-defined categories).

---

### Official Review · Reviewer_9PVb · 2023-11-02

**Soundness:** 2 fair
**Presentation:** 2 fair
**Contribution:** 1 poor
**Rating:** 3
**Confidence:** 4

**Summary:**

The paper proposes a hyperbolic architecture for sequential self-attention next-item recommendation applied to SASRec. Specifically, they replace the output prediction layer in SASRec with predictors in hyperbolic space. Besides, they adjust the machine precision setting to obtain a more accurate estimation of hyperbolic space curvature. Experimental results demonstrate that HSASRecCE can outperform SASRec with small embeddings.

**Strengths:**

1. The paper is the first work to extend the sequential self-attention next-item recommendation architecture by the hyperbolic prediction output layer.
2. The paper reveals that negative sampling harms the performance of hyperbolic models.
3. The paper adjusts the machine precision setting to obtain a more accurate estimation of hyperbolic space curvature, which can measure the hyperbolicity of a dataset and improve the performance of the hyperbolic model.

**Weaknesses:**

1. Limited novelty. The work simply replaces the output layer of SASRec with hyperbolic prediction layers. Besides, the used hyperbolic prediction layers(hyperbolic hyperplane and MLR) have been widely applied in other works. The contribution of this article needs to be re-condensed.
2. The paper measures the hyperbolicity of the dataset simply by δ-hyperbolicity but does not clarify the inherent hyperbolicity of the recommendation dataset. The utilization of hyperbolic space in this setting is questionable.
3. The paper does not explain and analyze why negative sampling harms the performance of the hyperbolic model, which does not occur in the Euclidean model.
4. The paper does not optimize the model with the compatibility of data with hyperbolic space, though it tries to obtain a more accurate estimation of the curvature of hyperbolic space. The role of curvature is not fully used in the recommendation design.
5. The authors state to apply hyperbolic geometry in the sequence learning settings. However, the applicability of the proposed strategy to the state-of-the-art recommendation models is questionable.
6. The experimental results are not convincing. The comparison seems to focus on PureSVD-N and EASEr from 2019 (4 years ago). It is a bit misleading since many works have emerged in the past 4 years (see [1, 2] for example).

[1] Zhou, Kun, et al. "Filter-enhanced MLP is all you need for sequential recommendation." Proceedings of the ACM web conference 2022. 2022.

[2] Xie, Xu, et al. "Contrastive learning for sequential recommendation." 2022 IEEE 38th international conference on data engineering (ICDE). IEEE, 2022.

**Questions:**

Please refer to Weaknesses.

**Details Of Ethics Concerns:**

No ethical issues.

---

> ### Author Response · Authors · 2023-11-22
> **Response to Reviewer 9PVb**
>
> We thank the reviewer for the thorough assessment and extensive feedback on our work. Below we address the outlined weakness points by quoting them and expanding the explanation below the quote. Due to space constraints, our response will be split into several comments below.
>
> ### Point 1
> > Limited novelty. The work simply replaces the output layer of SASRec with hyperbolic prediction layers. Besides, the used hyperbolic prediction layers(hyperbolic hyperplane and MLR) have been widely applied in other works. The contribution of this article needs to be re-condensed.
>
> We understand the concern and acknowledge that assessing novelty can be subjective. We believe it's important to note that the journey toward what might appear as a straightforward solution often involves intricate paths and complex problem-solving. While our final results may seem straightforward, achieving them involved rigorous exploration, refinement, and iterative processes that might not be immediately evident.
>
> It's also important to highlight the contextual specificity and the absence of a universally applicable framework for employing hyperbolic models. The application of the hyperbolic geometry in different domains often demands tailored adaptations and nuanced implementations. What might appear as a widely utilized approach in certain studies may not seamlessly translate into another problem set due to the intricacies and contextual nuances inherent in each domain. For example, we conducted several experiments on text-related tasks with the same approach and no immediate improvement was obtained there.
>
> Moreover, our contribution does not limit to the architecture design only. We provide additional experimentally-verified insights into hyperbolic geometry properties with respect to both models and data. We therefore are confident that the approach and methodology we applied in this task provide a nuanced perspective that adds value to the field.
>
> ### Point 2
>
> > The paper measures the hyperbolicity of the dataset simply by δ-hyperbolicity but does not clarify the inherent hyperbolicity of the recommendation dataset. The utilization of hyperbolic space in this setting is questionable.
>
> We totally agree with this viewpoint. Questioning the applicability of hyperbolic geometry is totally reasonable. Which is exactly why we focus a lot on providing additional context on the curvature estimation and compatibility of datasets. This is something that is largely absent in the majority of prior works. This void in understanding and analysis makes it harder to advance the field.
>
> We are not sure what the reviewer meant by the "inherent hyperbolicity of the recommendation dataset". To clarify our approach, measuring Gromov's $\delta$-hyperbolicity is the commonly used standard for understanding geometry of data in practical applications. By definition, it measures the extent to which a manifold deviates from the Euclidean geometry in favor of hyperbolicity. When applied to the recommendation dataset, one can say that it is supposed to uncover the inherent hyperbolicity of data.
>
> On a higher level, the question on whether the dataset is plausible for analysis via the hyperbolic geometry was answered in the seminal work by Krioukov et al. (2010). In that sense, we rely on the common understanding. We acknowledge though, that at least in our experiments the practice diverges from the theory. The emergence of "bad" examples may indicate a blind-spot in the conceptualization of the task via geometric lens or be an artifact of some undiscovered factors. One of our working hypotheses (yet to be tested) on the difference between "good" and "bad" datasets in terms of their hyperbolicity estimation is that the latter ones exhibit more heterogeneous structure. For example, the _Movielens-1M_ dataset, on which the hyperbolic models achieve excellent results, is more homogeneous. This is a "pure" domain of movies, and the items are more likely to coherently gather onto a well-defined manifold. Conversely, the _Amazon Arts Crafts and Sewing_ dataset seem to be more heterogenous with possibly many substructures, ingraining several different manifolds of different properties (e.g., different space curvatures), which prevents the models from learning good data representations. Verifying this hypothesis requires more elaborate analysis, which goes beyond the scope of the current work and presents an important direction for further research.

---

> > ### Author Response · Authors · 2023-11-22
> > **Response to Reviewer 9PVb (continued)**
> >
> > ### Point 3
> >
> > > The paper does not explain and analyze why negative sampling harms the performance of the hyperbolic model, which does not occur in the Euclidean model.
> >
> > We agree that such analysis would benefit community and advance understanding of applying hyperbolic geometry in machine learning tasks. This analysis links to the problem of finding connections of intrinsic structure of data to the properties of hyperbolic manifolds.
> >
> > While we do not provide an extensive analysis, we do suggest an initial  hypothesis in the last paragraph of Section 6.2, that links the popularity-induced hierarchical structure to the ability of hyperbolic models to capture distributional properties of data.
> >
> > Nevertheless, conducting a comprehensive analysis to verify the root causes of this issue presents a considerable challenge. We are unaware of a theoretical framework that helps conducting such investigations on a deeper level and would appreciate hints or direct references from the reviewer on that topic. To the best of our knowledge, we are the first to highlight that such a problem even exists.
> >
> > On the experimental front, we observed the degradation effect not only within our proposed model but also in our attempts to replicate [1], which replaces scalar products with distances and employs negative sampling for computational feasibility. However, these empirical observations fall short of uncovering the underlying nature of this effect; a robust methodology must be developed for a deeper understanding. Considering such a complexity of the task, we believe that addressing the issue of a strong impact of negative sampling on hyperbolic models deserves a dedicated research, that goes beyond the scope of the current work.
> >
> > [1] Vinh Tran, Lucas, Yi Tay, Shuai Zhang, Gao Cong, and Xiaoli Li. "Hyperml: A boosting metric learning approach in hyperbolic space for recommender systems." In _Proceedings of the 13th international conference on web search and data mining_, pp. 609-617. 2020.
> >
> > ### Point 4
> > > The paper does not optimize the model with the compatibility of data with hyperbolic space, though it tries to obtain a more accurate estimation of the curvature of hyperbolic space. The role of curvature is not fully used in the recommendation design.
> >
> > We respectfully disagree with the conclusion that curvature is not fully used in the recommendation design. The curvature parameter $c$ is involved in all transformations and algebraic operations related to the hyperbolic space in our approach, e.g., equations (5), (6) that are defined on a manifold with the set curvature $c$.  Moreover, the effects of varying values of $c$ on the quality of recommendations is demonstrated in Figure 4.
> >
> > ### Point 5
> > > The authors state to apply hyperbolic geometry in the sequence learning settings. However, the applicability of the proposed strategy to the state-of-the-art recommendation models is questionable.
> >
> > We respectfully disagree with this statement as well. The use of full cross-entropy  (CE) loss was recently shown to significantly boost the quality of the original SASRec model, please see the Klenitskiy & Vasilev (2023). As the authors demonstrate, simply switching from BCE to CE loss allows the SASRec architecture to outperform even more elaborate models such as BERT4Rec. In our work, we experiment with both CE and BCE variants of SASRec and show that the hyperbolic CE-based version outperforms its competitors. We note that the majority of recent works compare their models to the weaker BCE variant of the SASRec, not to the more capable CE version. Notably, like in the case with BERT4Rec, many of these works also use the CE loss (possibly extended with additional components), which may fully explain their advantage over the BCE-based version of SASRec. Unfortunately, none of these works provide an ablation study demonstrating isolated effects of using CE  loss vs adding other componets to the loss on the final quality. We would appreciate if the reviewer could point us to any such ablation studies in the case we missed them.
> >
> > We also acknowledge that it raises the question on a general utility of our approach to other architectures that extend the CE loss with additional components. Given the intricacies of adopting hyperbolic geometry to perform non-trivial operations, we believe that analysis of the compatibility of hyperbolic geometry with such extensions requires a thorough work and deserves a dedicated research that goes beyond the scope of the current work.

---

> ### Author Response · Authors · 2023-11-22
> **Response to Reviewer 9PVb (continued)**
>
> ### Point 6
>
> > The experimental results are not convincing. The comparison seems to focus on PureSVD-N and EASEr from 2019 (4 years ago). It is a bit misleading since many works have emerged in the past 4 years (see [1, 2] for example).
>
> We partially addressed this concern in the answer to p.5. Our main comparison target is the SASRec extension with CE loss that shows state-of-the-art performance. The referenced by the reviewer MLP model [1] compares to both SASRec (with BCE) and BERT4Rec. As we already stated above, the CE version of SASRec outperforms both of these baselines, and our hyperbolic variant of CE-based SASRec improves the metrics further.
>
> We acknowledge that comparing  with MLP model can be interesting. However, the design of this model relies on pairwise ranking loss that also utilizes the negative sampling. As we previously discussed, this sampling may not play well with hyperbolic geometry and we'd argue that final comparison with such models should be performed after the comprehensive analysis of the root cause of the negative sampling impact on hyperbolic models. It links us back to p.3, which we also addressed.
>
> The contrastive learning model from [2] compares only to the weaker BCE-based SASRec and no comparison is made with CE-based BERT4Rec. Overall, the relative improvement in this work looks weaker than the improvement of the full CE-based SASRec studied by Klenitskiy & Vasilev (2023). For example, on the Movielens-1M dataset in [2] one has:
> - SASRec's HR@10 = 0.1902,
> - CL3SRec's HR@10 = 0.1975,
> - the **relative improvement 3.8%**.
>
> The same dataset in Klenitskiy & Vasilev (2023):
> - BCE SASRec HR@10 = 0.2500,
> - CE SASRec HR@10 = 0.3152,
> - the **relative improvement 26%**.
>
> The experimental setup differs a bit in these works. However, the resulting difference in the absolute values of the metrics is not significant, while the relative improvement over the original BCE-based SASRec baseline is much more pronounced in Klenitskiy & Vasilev (2023). This allows us to conclude that [2] provides a weaker baseline than the CE-based SASRec.
>
> Additionally, the same remark on negative sampling that we did w.r.t MLP model holds for [2] as well. We envision that CL3SRec won't gain much from the hyperbolic geometry due to the negative sampling utilized in part of its loss. The effects of switching to full CE loss in this case are more intricate, as CL3SRec uses a composite multi-task learning. Adopting hyperbolic geometry would require additional efforts to properly combine all parts of the loss. Performing such work would significantly extend our current approach and make it hard to fit into a single conference paper.

---

### Official Review · Reviewer_vqza · 2023-11-05

**Soundness:** 2 fair
**Presentation:** 2 fair
**Contribution:** 2 fair
**Rating:** 3
**Confidence:** 5

**Summary:**

This paper presents an approach that leverages hyperbolic geometry for extending recommendation systems. The idea is that the hyperbolic layer will capture structural properties of the approach. The authors are based on one base model, SASRec, which they extend. They perform experiments on multiple datasets with mixed results.

**Strengths:**

The paper is generally well written which I greatly appreciate. It contains some interesting ideas of how to extend recommendation systems to take advantage of hyperbolic geometries. There is quite good coverage of related work even though I would like to see references from practical work. The authors could check recent papers in recommender systems from Pinterest, Airbnb, etc. where there are also real use-cases.

**Weaknesses:**

My first comment is that the originality of this work is quite limited as other works proposed hyperbolic recommenders. The addition of this paper is the extension of SASRec with a hyperbolic layer as stated at the last sentence Section 5. I am not sure how much originality is there.

As mentioned previously, it would be great if the authors could add some references from recent works of recommender systems on real use-cases just to contrast it with the current SOTA in real applications. Do we expect that such an approach would be viable or would give better online results in a real system?

In Section 3.2 authors mention that negative sampling is typically implemented using uniform sampling which is rarely the case for real use cases where one employs heuristics in order to sample hard negatives and learn more robust models.

I do not understand the statement "distributional properties of items based on the popularity hierarchy". What exactly someone would like to capture here? And why popularity cannot be captured with non-hyperbolic geometry?  I cannot see how this would help a recommendation system.

In the experimental part I would add the following baselines:
- Most popular
- Co-occurence baseline, where one would recommend the property that co-occurs more often with the last item in the sequence. This is often a very strong baseline.

Adding challenging datasets also would be great. In recent years many datasets have been released that can be used for sequential recommendation. For example:
- https://xmrec.github.io/wsdmcup/
- https://github.com/ExpediaGroup/pkdd22-challenge-expediagroup

Generally the results are no convincing/great. It seems in some cases there is some slight improvement (third digit). And the classification in good and bad datasets is arbitrary. What is the hypothesis that in one dataset of Amazon data the hyperbolic version is better and in another one (like Products) is not better? This seems random.

**Questions:**

Please see weaknesses section for the questions.

---

> ### Author Response · Authors · 2023-11-22
> **Response to Reviewer vqza**
>
> We are grateful to the reviewer for their careful consideration of our work and for the provided suggestions on possible improvements. Below we address the outlined concerns by quoting them and expanding the explanation below the quote. Due to space constraints, our response will be split into several comments below.
>
> ### Point 1
> > My first comment is that the originality of this work is quite limited as other works proposed hyperbolic recommenders. The addition of this paper is the extension of SASRec with a hyperbolic layer as stated at the last sentence Section 5. I am not sure how much originality is there.
>
> We recognize that at a superficial glance, incorporating a hyperbolic layer into the SASRec framework may appear as an incremental addition to the existing body of work. However, it is imperative to emphasize the intricate and meticulous process involved in this extension. Our approach involved a nuanced exploration of hyperbolic geometry properties with respect to integration within the sequential learning architectures. The final solution was selected among several competing one and required extensive experimentation. Moreover, our contribution does not limit to the architecture design only. We provide additional empirical insights into hyperbolic geometry properties with respect to both models and data. We therefore are confident that the approach and methodology we applied in this task provide a nuanced perspective that adds value to the field and highlights important issues that inform new directions for further research.
>
> ### Point 2
> > As mentioned previously, it would be great if the authors could add some references from recent works of recommender systems on real use-cases just to contrast it with the current SOTA in real applications. Do we expect that such an approach would be viable or would give better online results in a real system?
>
> We appreciate the reviewer's suggestion and recognize the value in contrasting our approach with recent works on real-world recommender systems. We note that our work builds upon the research by Klenitskiy and Vasilev (2023). According to the stated affiliations, the research is conducted within an AI Lab of a large company and provides insights into the ways of significantly improving an already strong sequential learning model. Consequently, our work suggests further improvements to the demonstrated state-of-the-art results. Our approach offers a better quality of recommendations with more compact representation at the same time. We believe, it therefore holds significant practical implications worth sharing with the community.
>
> Our paper delineates the strengths and limitations of our proposed approach's applicability. Furthermore, we offer the complete codebase, enabling straightforward reproduction and adaptation of our solution to specific real-world applications. This comprehensive sharing empowers practitioners to evaluate the practicality of our method, considering its merits and shortcomings in their individual contexts.
>
> It's crucial to underscore that the realm of recommender systems lacks a universal "silver bullet" solution applicable across all domains, and there's an overall agreement in the community that developing a silver bullet is not the end goal [1]. Assessing the viability of a particular model within a specific use case traverses subjective terrain, limiting a scientific approach and potentially harming the field's overall progress.
>
> [1] Ferrari Dacrema, Maurizio, Simone Boglio, Paolo Cremonesi, and Dietmar Jannach. "A troubling analysis of reproducibility and progress in recommender systems research." _ACM Transactions on Information Systems (TOIS)_ 39, no. 2 (2021): 1-49.

---

> > ### Author Response · Authors · 2023-11-22
> > **Response to Reviewer vqza (continued)**
> >
> > ### Point 3
> > > In Section 3.2 authors mention that negative sampling is typically implemented using uniform sampling which is rarely the case for real use cases where one employs heuristics in order to sample hard negatives and learn more robust models.
> >
> > While improving the negative sampling can make the original SASRec model work better, our main comparison target was the much stronger CE-based model that doesn't depend on the negative sampling. BCE is only served as an approximation to full CE loss, so the no-sampling model presents the strongest baseline.
> >
> > On the other hand, we acknowledge the importance of additional analysis of the effects that connect negative sampling strategies to the performance of the hyperbolic models. We believe that exploring such effects should be linked to the better analysis of the connections between the structural properties of data and geometry. We expand on this issue a bit further in p.3 in the comments under the Reviewer 9PVb review. Conducting a comprehensive analysis to verify the root causes of the described sampling issue presents a considerable challenge. We are unaware of a theoretical framework that helps conducting such investigations on a deeper level and would appreciate hints or direct references from the reviewer on that topic. To the best of our knowledge, we are the first to highlight that such a problem even exists.
> >
> > ### Point 4
> > > I do not understand the statement "distributional properties of items based on the popularity hierarchy". What exactly someone would like to capture here? And why popularity cannot be captured with non-hyperbolic geometry? I cannot see how this would help a recommendation system.
> >
> > Thank you for bringing this question up. We agree that this can be beneficial to expand a bit further on this matter without cutting the corners. As outlined in Krioukov et al. (2010), hyperbolic geometry is suitable for embedding hierarchies. For example, one of the well-known results is that one can embed an arbitrary tree into a 2D Poincaré disk [2]. Closer to the disk boundary, the shortest path between two points is almost the same as the path through the origin, which resembles the properties  of graphs, where the shortest path between two child nodes goes through the parent node [3]. In the case of recommender systems, the most straightforward candidate for representing hierarchies is the popularity-induced distribution of items. That is what we meant by "distributional properties of items based on the popularity hierarchy". In other words, one of the potential advantages of the hyperbolic geometry is that it can help to learn better representation of data that contains some sort of hierarchical structure, which in our case is related to different popularity levels of items. This also aligns well with the general analysis from Krioukov et al. (2010) stating that hyperbolic geometry is especially suitable for modeling complex networks that by definition follow non-trivial distribution, e.g, power-law degree distribution. The standard collaborative filtering datasets are known to exhibit such properties, i.e., items distribution in them can be accurately described by the power-law or zipf-law. So, generally, while popularity-induced hierarchy may can be captured in different ways, there's a theoretical ground for trying to capture it through the hyperbolic geometry, which seem to be the most suitable for this task. It largely motivated our research in the direction of hyperbolic geometry.
> >
> > [2] Rik Sarkar. "Low distortion delaunay embedding of trees in hyperbolic plane." In _International Symposium on Graph Drawing_, pages 355–366. Springer, 2011.
> > [3] Sala, Frederic, Chris De Sa, Albert Gu, and Christopher Ré. "Representation tradeoffs for hyperbolic embeddings." In _International conference on machine learning_, pp. 4460-4469. PMLR, 2018.

---

> ### Author Response · Authors · 2023-11-22
> **Response to Reviewer vqza (continued)**
>
> ### Point 5
> > In the experimental part I would add the following baselines:
> Most popular
> Co-occurence baseline, where one would recommend the property that co-occurs more often with the last item in the sequence. This is often a very strong baseline.
>
> We thank the reviewer for the suggestion of additional baselines. In the case of the last-item-cooccurrence model, we agree that it may provide a reasonable quality. It would however add no additional understanding into the difference between Euclidean and hyperbolic sequential learning models, which is one of the main targets in our work.
>
> We do have results for the popularity-based model, though, listed below. One can see that the performance is very poor, which is not surprising.
> - Movielens-1M
> 	- NDCG@10  0.0214
> 	- HR@10    0.0427
> 	- MRR@10   0.0151
> 	- COV@10   0.0529
> - Grocery and Gourmet Food
> 	- NDCG@10  0.0256
> 	- HR@10    0.0336
> 	- MRR@10   0.0230
> 	- COV@10   0.0005
> - Office Products
> 	- HR@10    0.0074
> 	- MRR@10   0.0024
> 	- NDCG@10  0.0036
> 	- COV@10   0.0006
> - Arts Crafts and Sewing
> 	- NDCG@10  0.0096
> 	- HR@10    0.0150
> 	- MRR@10   0.0080
> 	- COV@10   0.0007
>
> We do not report these results to save space and avoid distracting the reader from the main effects.
>
> ### Point 6
> > Adding challenging datasets also would be great. In recent years many datasets have been released that can be used for sequential recommendation.
>
> We thank the reviewer for the additional references to rich datasets. This can be a great addition for the deeper analysis of the data compatibility, once the framework for such analysis is developed.

---

> ### Author Response · Authors · 2023-11-22
> **Response to Reviewer vqza (continued)**
>
> ### Point 7
> > Generally the results are no convincing/great. It seems in some cases there is some slight improvement (third digit). And the classification in good and bad datasets is arbitrary. What is the hypothesis that in one dataset of Amazon data the hyperbolic version is better and in another one (like Products) is not better? This seems random.
>
> We respectfully disagree with the assessment regarding the improvement observed. Notably, the improvements were achieved utilizing  _smaller embedding sizes_. Please, see Table 4 in the Appendix for more detailed breakdown. Our results showcase that even if the metrics were to remain at par without any explicit improvement over the baseline, our approach would still offer an advantage in terms of _more compact representation_ while maintaining the state-of-the-art quality. This comparison underscores the practical advantage inherent in our approach.
>
> In terms of the categorization of the datasets, we also recognize the importance of further in-depth analysis to understand what aspects render datasets more compatible with hyperbolic geometry. However, as we discussed in the response to p.2 and p.3 of the Reviewer 9PVb comments, there are several considerable challenges associated with the task.  In the absence of better analysis tools that are yet to be invented, we provide an empirical observation and link it to the most visible differences in the $\delta$-hyperbolicity estimation. It is indeed not suitable for uncovering the deep underlying effects, but it is not random.
>
> There seem to be no established theoretical framework for a deeper analysis of data compatibility. We acknowledge that the practice diverges from the theory in our experiments (a circumstance often encountered in general). The theory prescribes that hyperbolic geometry is suitable for any instance of complex networks family, which collaborative filtering datasets  belongs to. The emergence of "bad" practical examples (for which the estimation of $\delta$-hyperbolicity is not reliable) may indicate a blind-spot in the conceptualization of the task via geometric lens or be an artifact of some undiscovered factors.
>
> One of our working hypotheses (pending further validation) regarding the distinction between "good" and "bad" datasets in their hyperbolicity estimation is that the latter tends to display a more heterogeneous structure. For example, the _Movielens-1M_ dataset, on which the hyperbolic models achieve excellent results, is more homogeneous. This is a "pure" domain of movies, and the items are more likely to coherently gather onto a well-defined manifold. Conversely, the _Amazon Arts Crafts and Sewing_ dataset seem to be more heterogenous with possibly many substructures encompassing different types of products or consumption patterns, therefore ingraining several different manifolds of different properties (e.g., different space curvatures). This in turn may prevent hyperbolic  models from learning better data representations. Verifying this hypothesis requires more elaborate analysis and presents an important direction for further research.
>
> It is worth adding an additional remark on the quality improvement here, that we also made in response to p.2 of the Reviewer WdM2.  We deliberately restrict hyperbolic models to the reduced embedding sizes.  If we lift this restriction, even on "bad" datasets the hyperbolic models can be trivially made to perform at the same level as their Euclidean counterparts. It simply takes setting higher values of $\delta$ (lower curvature $c$ respectively). In this case the hyperbolic models become practically indistinguishable from the Euclidean ones. This result, however, would prevent us from demonstrating the true behavior of hyperbolic models on problematic datasets, while making the formal results (quality metrics) look much better in comparison. We believe that this result would be too trivial to show and instead focus on the discrepancies in learning abilities depending on the geometric properties of data. We believe that our approach offers a more equitable perspective, helping to more effectively unveil the underlying intricacies associated with training hyperbolic models.

---

### Meta-Review · Area_Chair_HJmx · 2023-12-14

**Metareview:**

The paper extends SASRec with hyperbolic geometry, specifically replacing the output prediction layer in SASRec with predictors in hyperbolic space and showed improved results in next-item prediction.

Strength: the proposed method is straight-forward, and easy to follow. The proposed method was able to achieve similar performance to the Euclidean model with fewer dimensions. The authors made several meticulous designs, e.g., curvature estimation, which lead to the success.
Weakness: the paper combines several existing technologies for recommendation use cases. All the reviewers raised concerns on the novelty of the method and asked for more insights to shed lights on the underlying mechanism of the proposal. The paper will enjoy better audience in a RecSys community.

**Justification For Why Not Higher Score:**

The paper combines several existing technologies for recommendation use cases, although made several meticulous designs, which probably made it work. The reviewers raised concerns on the novelty of the method and asked for more insights to shed lights on the underlying mechanism of the proposal. The paper will enjoy better audience in a RecSys community.

**Justification For Why Not Lower Score:**

N/A

---

### Decision · Program_Chairs · 2024-01-16

Reject